# Low Levels of TRIM28-Interacting KRAB-ZNF Genes Associate with Cancer Stemness and Predict Poor Prognosis of Kidney Renal Clear Cell Carcinoma Patients

**DOI:** 10.3390/cancers13194835

**Published:** 2021-09-28

**Authors:** Patrycja Czerwinska, Andrzej Adam Mackiewicz

**Affiliations:** 1Department of Cancer Immunology, Poznan University of Medical Sciences, 15 Garbary St., 61-866 Poznan, Poland; andrzej.mackiewicz@wco.pl or; 2Department of Diagnostics and Cancer Immunology, Greater Poland Cancer Centre, 15 Garbary St., 61-866 Poznan, Poland

**Keywords:** KRAB-ZNF, TRIM28, cancer stemness, TCGA, KIRC

## Abstract

**Simple Summary:**

This is the first report investigating the involvement of TRIM28-interacting KRAB-ZNFs in kidney cancer progression. We demonstrate a significant negative association between KRAB-ZNFs and cancer stemness followed by an attenuated immune-suppressive response and reveal the prognostic role for several KRAB-ZNFs. Our findings may help better understand the molecular basis of kidney cancer and ultimately pave the way to more appropriate prognostic tools and novel therapeutic strategies directly eradicating the dedifferentiated compartment of the tumor.

**Abstract:**

Krüppel-associated box zinc finger (KRAB-ZNF) proteins are known to regulate diverse biological processes, such as embryonic development, tissue-specific gene expression, and cancer progression. However, their involvement in the regulation of cancer stemness-like phenotype acquisition and maintenance is scarcely explored across solid tumor types, and to date, there are no data for kidney renal clear cell cancer (KIRC). We have harnessed The Cancer Genome Atlas (TCGA) and the Gene Expression Omnibus (GEO) database transcriptomic data and used several bioinformatic tools (i.e., GEPIA2, GSCALite, TISIDB, GSEA, CIBERSORT) to verify the relation between the expression and genomic alterations in KRAB-ZNFs and kidney cancer, focusing primarily on tumor dedifferentiation status and antitumor immune response. Our results demonstrate a significant negative correlation between KRAB-ZNFs and kidney cancer dedifferentiation status followed by an attenuated immune-suppressive response. The transcriptomic profiles of high KRAB-ZNF-expressing kidney tumors are significantly enriched with stem cell markers and show a depletion of several inflammatory pathways known for favoring cancer stemness. Moreover, we show for the first time the prognostic role for several KRAB-ZNFs in kidney cancer. Our results provide new insight into the role of selected KRAB-ZNF proteins in kidney cancer development. We believe that our findings may help better understand the molecular basis of KIRC.

## 1. Introduction

The largest class of DNA-binding transcription factors in mammalian cells, with more than 400 genes encoding for at least 700 members, is known as the Krüppel-associated box zinc finger (KRAB-ZNF) family [1]. KRAB-ZNF proteins contain tandem copies of the C_2_H_2_ zinc finger DNA binding motif (ZNFs) accompanied with conserved Krüppel-associated box (KRAB) domain, which mediates the recruitment of a specific and universal corepressor—KRAB-associated protein 1 (KAP1), also known as tripartite-motif containing 28 (TRIM28) or transcriptional intermediary factor 1β (TIF1β) [2]. In humans, all KRAB-ZNFs studied in detail are repressors and mostly utilize the KRAB domain to bind the TRIM28 corepressor [3]. TRIM28 is indispensable for transcriptional repression and gene silencing, acting as a scaffold protein for various heterochromatin-inducing factors, including histone deacetylases (nucleosome remodeling deacetylase (NuRD), histone methyltransferases (SET domain bifurcated 1 (SETDB1)), and heterochromatin proteins (heterochromatin protein 1 (HP1)). These proteins modify chromatin structure, leading to a compacted, silent state (epigenetic repression) [4].

Generally, KRAB-ZNFs are sequence-specific DNA binding proteins that bind TRIM28, further recruiting complex epigenetic machinery, resulting in reversible repression of target gene transcription [5]. However, to date, the direct interaction with TRIM28 has been confirmed experimentally only for several KRAB-ZNFs.

KRAB-ZNFs are known to regulate diverse biological processes, such as embryonic development, tissue-specific gene expression, and cancer progression, although their role in kidney cancer development remains largely unknown [2,3,5,6]. Kidney renal clear cell carcinoma (KIRC) is characterized by substantial heterogeneity during tumor development, causing considerable challenges to precise treatment. Notably, the population of cancer cells with stem cell-like characteristics might facilitate tumor heterogeneity [7].

It is postulated that cancer cells, at least partially, may experience phases of transition between stem-like and non-stem-like states, and epigenetic dysregulation might contribute to tumorigenicity via facilitating this phenomenon. Stem cell-associated molecular features of cancer cells are indisputably necessary for disease development and progression, endowing cancers with intrinsic resistance to standard therapies and providing tumor relapse after treatment [8,9]. 

Recent artificial intelligence-supported approaches have led to the development of so-called stemness indices that quantify tumor dedifferentiation status with great effectiveness. These molecular signatures represent an essential step in designing novel therapeutic schemes [10,11,12,13]. 

The involvement of KRAB-ZNFs in the regulation of cancer stemness-like phenotype acquisition and maintenance is scarcely explored across solid tumor types, and to date, there are no data for kidney renal clear cell cancer. Therefore, we have harnessed The Cancer Genome Atlas (TCGA) and the Gene Expression Omnibus (GEO) database transcriptomic data to verify the relation between the expression and genomic alterations in KRAB-ZNFs and kidney cancer patients’ survival, focusing primarily on tumor dedifferentiation status. 

Using the Compartmentalized Protein–Protein Interaction Database (ComPPI) [14] and Pathway Commons database [15], we filtered 28 KRAB-ZNFs that directly interact with TRIM28 and possess both KRAB and ZNF domains. 

Using the GSCALite platform [16], we demonstrated that 17 out of 28 KRAB-ZNFs are differentially expressed in KIRC tumor tissues compared to normal adjacent tissues and that most KRAB-ZNFs were correlated with each other in the tumor. According to the GEPIA2 database [17], for 18 KRAB-ZNFs, we reported a significant association of upregulated expression with KIRC TCGA cancer patients’ prolonged survival, suggesting their tumor-suppressive role. 

Furthermore, the expression of at least 20 KRAB-ZNFs was significantly associated with tumor dedifferentiation status (tumor grade and disease stage), mostly negatively, which was further confirmed with cancer stemness scores. Moreover, the gene set enrichment analysis (GSEA) [18] of gene expression profiles associated with specific KRAB-ZNFs revealed their significant depletion with stem cell markers in KIRC, further supporting our first observation. 

Next, we constructed a potential prognostic KRAB-ZNF-based signature and confirmed its significance in predicting the disease course of patients with KIRC. The gene expression profiles of high-risk KIRC patients were significantly enriched with cancer stemness markers and immune infiltration-associated gene sets. 

Accordingly, using a CIBERSORT algorithm [19], we demonstrated that the expression of most tested KRAB-ZNFs is negatively correlated with the level of regulatory T cells (Tregs), follicular helper T (Tfh) cells, and gamma/delta T (γδ T) cells. On the other hand, the abundance of resting mast T cells and resting NK cells was higher in high KRAB-ZNF-expressing KIRC tumors. Furthermore, increased expression of KRAB-ZNFs is significantly associated with the downregulation of many proinflammatory signaling pathways in KIRC, especially those mediated by interferons, IL2, IL6, or TNF-α, which was previously shown to enhance cancer stemness maintenance.

This is the first report that investigates the expression of TRIM28-interacting KRAB-ZNFs in kidney cancer. Our results demonstrate a significant negative correlation between KRAB-ZNFs and kidney cancer dedifferentiation status followed by an attenuated immune-suppressive response. Moreover, we show for the first time the prognostic role for several KRAB-ZNFs in kidney cancer. We believe that our findings may help better understand the molecular basis of KIRC and ultimately pave the way to more appropriate prognostic tools for KIRC and facilitate the development of new therapeutic strategies directly targeting the dedifferentiated compartment of the tumor.

## 2. Materials and Methods

### 2.1. Kidney Renal Clear Cell Carcinoma Data from TCGA and GEO Databases

In the current study, we used the transcriptomic data of kidney renal clear cell cancer (KIRC) from the TCGA database [20] and transcriptomic data from the GSE2109 and GSE11024 [21] gene sets from the GEO database. All data are available online, and the access is unrestricted and does not require patients’ consent or other permissions. The use of the data does not violate the rights of any person or any institution.

### 2.2. Identification of TRIM28-Interacting KRAB-ZNFs

To search for TRIM28-interacting KRAB-ZNFs, we used (i) the Compartmentalized Protein–Protein Interaction Database (ComPPI; v2.1.1; https://comppi.linkgroup.hu/protein_search, accessed on 12 July 2021), which provides qualitative information on the interactions, proteins, and their localizations integrated from multiple databases [14], and (ii) the Pathway Commons database (https://www.pathwaycommons.org/, accessed on 12 July 2021), an integrated resource of publicly available information about biological pathways, including biochemical reactions; assembly of biomolecular complexes; transport and catalysis events; and physical interactions involving proteins, DNA, RNA, and small molecules (e.g., metabolites and drug compounds) [15]. KRAB-ZNFs selected for our study are summarized in Appendix A.

### 2.3. The Expression of KRAB-ZNF Family Members in Distinct TCGA Cohorts

The expression of selected KRAB-ZNF family members in tumor tissues and normal adjacent tissues in KIRC tumors was analyzed using the Expression module of the GSCALite (http://bioinfo.life.hust.edu.cn/GSCA/#/, accessed on 13 September 2021) [16]. The module allows users to study the differential expression (based on normalized RSEM mRNA expression) between tumor and adjacent normal tissues for any gene of interest across 14 TCGA tumor types. The statistical significance was estimated by a *t*-test.

### 2.4. The Association between KRAB-ZNF Family Members’ Expression and Patients’ Outcome

The association between KRAB-ZNF family members’ expression and patients’ overall survival (OS) in TCGA KIRC tumor patients was analyzed with the Survival_Analysis panel of the GEPIA2 database (http://gepia2.cancer-pku.cn/#index, accessed on 15 July 2021) [17]. Similarly, the survival of kidney renal papillary cell carcinoma (KIRP) and kidney chromophobe carcinoma (KICH) TCGA patients was correlated with the level of tested KRAB-ZNFs using the GEPIA2 database.

### 2.5. TCGA Genomic Data

Genomic data for TCGA KIRC tumors were directly downloaded from the cBioPortal (www.cbioportal.org, accessed on 16 July 2021) database [22].

### 2.6. Transcriptomic Data

The RNA sequencing-based mRNA expression data were directly downloaded from the cBioPortal. RNASeq V2 from TCGA is processed and normalized using RSEM [23]. Specifically, the RNASeq V2 data in cBioPortal corresponds to the rsem.genes.normalized_results file from TCGA. The Spearman’s correlation was used for detection of coexpressed genes with *p*-value < 0.05 and FDR < 0.01 as cut-offs. Differentially expressed genes (DEGs) were cut off at *p*-value < 0.05 and FDR < 0.05.

### 2.7. Stemness-Associated Scores

The mRNA-SI stemness score [10] and other stemness signatures (Ben-Porath_ES_core, Wong_ESC_core, Bhattacharya) used in this study were previously described [11,12,13].

### 2.8. Histologic Tumor Grade and Stage

The association between KRAB-ZNF family members’ expression and the histologic tumor grade or stage was assessed using the TISIDB portal (http://cis.hku.hk/TISIDB/index.php, accessed on 20 July 2021) [24]. The correlations were calculated using Spearman’s rank correlation coefficient (r).

### 2.9. Gene Set Enrichment Analysis

The Gene Set Enrichment Analysis (GSEA, http://www.broad.mit.edu/gsea/index.html, accessed on 29 July 2021) [18] was used to detect the coordinated expression of a priori defined groups of genes within the tested samples. Gene sets are available from the Molecular Signatures Database (MSigDB, http://www.broad.mit.edu/gsea/.msigdb/msigdb_index.html, accessed on 29 July 2021). All significantly correlated genes (Spearman correlation, FDR < 0.01) were imported to GSEA. The GSEA was run according to the default parameters: each probe set was collapsed into a single gene vector (identified by its HUGO gene symbol), permutation number = 1000, and permutation type = “gene-sets.” The FDR < 0.01 was used to correct for multiple comparisons and gene set sizes.

### 2.10. Construction of a Potential Prognostic Signature

Preselected KRAB-ZNFs were fitted in a univariate and multivariate Cox proportional hazards regression analysis (Appendix A). Risk scores were estimated by a formula that combines the expression levels of KRAB-ZNFs weighted by their estimated regression coefficients in the multivariate Cox regression model. The exact formula was as follows: risk score  =  (βZNF2 × expression level of ZNF2)  +  (βZNF256 × expression level of ZNF256)  +  (βZNF382 × expression level of ZNF382) + (βZNF420 × expression level of ZNF420) + (βZNF496 × expression level of ZNF496) + (βZNF585B × expression level of ZNF585B) + (βZNF829 × expression level of ZNF829). Patients were divided into high-risk and low-risk groups using the median risk score as a cut-off. The differences in patient survival between groups were estimated by the Kaplan–Meier survival analysis and log-rank (Mantel–Haenszel) test. The receiver operating characteristic (ROC) curve for the risk score and survival status (0—deceased, 1—living) was plotted in GraphPad Prism 8.0 software (GraphPad Software, Inc., La Jolla, CA, USA) to assess the predictive accuracy of a prognostic model. 

### 2.11. Immune-Associated Scores, CIBERSORT, and Immune KIRC Subtypes

The leukocyte fraction and the lymphocyte infiltration signature (LIS) scores were estimated based on the transcriptome profiles of each sample, as previously reported [25]. Infiltrating immune cell fractions in low- and high-risk KIRC patients were quantified according to the CIBERSORT algorithm [19]. Classification of KIRC samples into six distinct immune subtypes was previously reported [25].

### 2.12. Other Statistical Analyses

Statistical analyses were carried out with GraphPad Prism 8.0 software (GraphPad Software, Inc., La Jolla, CA, USA). Multiple comparisons were performed with the ANOVA test. The correlation between two variables was assessed with Spearman’s rank correlation coefficient (r).

## 3. Results

### 3.1. TRIM28-Interacting KRAB-ZNF Proteins Are Differentially Expressed in Tumor vs. Normal Kidney Tissue, and Their High Expression Is Associated with Better KIRC Patient Survival

Firstly, using the ComPPI database [14] and Pathway Commons database [15], we identified 28 KRAB-ZNF proteins that directly interact with TRIM28 protein (Appendix A). These proteins, namely ZFP1, ZFP57, ZNF2, ZNF10, ZNF74, ZNF133, ZNF140, ZNF195, ZNF197, ZNF224, ZNF250, ZNF256, ZNF264, ZNF274, ZNF324B, ZNF331, ZNF350, ZNF354A, ZNF382, ZNF420, ZNF460, ZNF496, ZNF585B, ZNF620, ZNF689, ZNF747, ZNF764, and ZNF829, were the objects of our study (Appendix A). 

Using the GSCALite database [16], we analyzed the expression of preselected KRAB-ZNFs in TCGA KIRC tumor vs. normal adjacent tissue. We observed that 10 members are significantly downregulated, while 7 members are significantly overexpressed in tumor tissue (Figure 1A and Appendix A). Similar results were observed in the additional GEO dataset (Mixed Renal, *n* = 79, GSE11024; Appendix A). We observed a significant positive correlation with each other for most of the tested KRAB-ZNFs in TCGA KIRC (Figure 1B). According to the cBioPortal database, all tested KRAB-ZNFs are rarely mutated in KIRC tumors, with only ZNF197, ZNF354A, and ZNF620 being altered in more than 10% of KIRC patients (Appendix A). 

Next, using the median expression as a cut-off, we observed that 18 KRAB-ZNFs are significantly associated with better patient survival in KIRC (Figure 1C and Appendix A) in contrast to other kidney tumors: kidney renal papillary cell carcinoma (KIRP) or kidney chromophobe cancer (KICH) (Appendix A), where no statistically significant relevance to patients’ outcome was detected. 

### 3.2. High Expression of TRIM28-Interacting KRAB-ZNFs Is Negatively Associated with Cancer Dedifferentiation Status, and Stemness Markers Are Significantly Depleted in KRAB-ZNF-Associated Transcriptome Profiles 

We further analyzed the association between KRAB-ZNF expression and tumor dedifferentiation status. Specifically, we looked at the level of preselected KRAB-ZNFs in KIRC regarding tumor stage and tumor grade and observed that most KRAB-ZNFs are negatively associated with both clinicopathological features (Figure 2A,B). Specifically, out of 28 tested, the expression of 16 and 20 KRAB-ZNFs is negatively correlated with the KIRC tumor stage and tumor grade, respectively.

As dedifferentiated tumors clearly exhibit stemness characteristics, we analyzed the association between KRAB-ZNF expression and previously defined transcriptome-based stemness scores, namely mRNA-SI [10], Ben-Porath_ES1 signature [11], Wong_ESC_core signature [12], and Bhattacharya_hESC signature [13] scores. For 21 KRAB-ZNFs, we observed a significant negative correlation with the level of at least two stemness signatures (Figure 2C). These data further confirm the negative association between KRAB-ZNF expression and tumor dedifferentiation status. 

Next, we defined the KRAB-ZNF-related transcription profiles as all markers significantly correlated (*p* < 0.05, FDR < 1%) with each of the tested KRAB-ZNFs in TCGA KIRC data (Appendix A). Then, we used the Gene Set Enrichment Analysis (GSEA) [18] to verify whether the KRAB-ZNF-associated transcription profiles are enriched with or depleted of stemness markers. As presented in Figure 2D, we observed significant depletion of stemness markers in almost all tested KRAB-ZNF-associated transcription profiles (Wong_ESC_core gene set), which was further validated with a different gene set attributed to cell stemness—Mueller_Plurinet [26] (Appendix A). Moreover, similar results were obtained for the additional GEO dataset (Tumor Kidney, *n* = 261, GSE2109), further confirming a negative association between KRAB-ZNF expression and cancer stemness (Appendix A).

### 3.3. KRAB-ZNF-Based Gene Signature Predicts the Prognosis of KIRC Patients

Next, to build a predictor model, TRIM28-associated KRAB-ZNFs were subjected to a univariate Cox regression model. In total, seven KRAB-ZNFs were significantly correlated with the overall survival of KIRC patients (Appendix A), exhibiting a negative coefficient that confirms that their higher expression is associated with prolonged survival. KRAB-ZNFs not prognostically relevant for overall survival (Cox univariate analysis *p* > 0.05) were omitted from further prognosis evaluation. The 516 TCGA KIRC patients were randomly divided into a discovery set (*n* = 258) and a validation set (*n* = 258). Based on the expression level of seven KRAB-ZNFs and multivariate Cox regression coefficients for a discovery set, we built a risk score formula (detailed in Section 2) for the prediction of KIRC patients’ survival (Appendix A), which was further validated in a validation set (Appendix A) and the entire KIRC cohort (Figure 3). 

As demonstrated in Figure 3A, the high-risk group suffered a worse prognosis than patients in the low-risk group (median OS: 71.5 months vs. undefined, respectively). The distribution of the risk score, patients’ survival status, and the expression of prognostic KRAB-ZNFs were ranked according to the risk score value (Figure 3B). Patients with a high risk score demonstrate greater mortality and lower expression of KRAB-ZNFs than patients in the low-risk group. Our risk score shows a predictive value with the area under the curve (AUC) equal to 0.656 in the discovery set, AUC equal to 0.666 in the validation set, and AUC equal to 0.661 in the entire KIRC cohort (Appendix A).

### 3.4. The Gene Expression Profiles of High-Risk Patients Are Enriched with Cancer Stemness Markers and Immune Infiltration-Associated Gene Sets

Next, we compared the gene expression profiles of high- and low-risk KIRC patients (Appendix A) and observed significant enrichment of immune response-associated terms followed by significant enrichment of stemness-associated biological terms in the high-risk KIRC cohort (Appendix A). We observed that our risk score was significantly correlated with the stemness-associated transcriptome-based signatures (mRNA-SI, *p* = 0.0139; Ben-Porath, *p* < 0.0001; Wong, *p* < 0.0001; Bhattacharya, *p* < 0.0001; Figure 4A,B). The GSEA analysis also confirmed significant enrichment of gene expression profiles of high-risk KIRC patients with previously defined stemness signatures (Figure 4C). Moreover, among the high-risk patients, the frequencies of higher-grade (Figure 4D) and higher stage (Figure 4E) tumors were significantly elevated (*p* = 6.91 × 10^−8^ and *p* = 4.838 × 10^−6^, respectively), suggesting that high-risk tumors exhibit dedifferentiated phenotype.

Next, we compared the level of transcriptome-based immune-related scores, namely the leukocyte fraction and the leukocyte infiltration signature (LIS) scores [25], with the level of our risk score and observed a significant positive correlation (Figure 5A,B). Higher-risk patients exhibit higher levels of both immune scores (Figure 5C,D). Among all tested 22 immune populations estimated by the CIBERSORT [19] (Appendix A), we observed a significant increase in regulatory T cells (*p* = 0.0009) (Figure 5E) and follicular helper T cells (*p* = 0.0014) (Figure 5F) followed by a decreased number of resting mast cells (*p* = 0.0371) (Figure 5G) in high-risk KIRC patients. Furthermore, among the high-risk KIRC patients, the frequencies of C1 (“wound healing”), C2 (“IFN-γ dominant”), and C6 (“TGF-β dominant”) immune subtypes were significantly elevated (*p* = 0.0054) when compared to low-risk KIRC cohort. Taken together, these results indicate that high-risk KIRC tumors exhibit a more stem cell-like phenotype with accumulated immune-suppressive cells and signaling.

### 3.5. The Association between KRAB-ZNF Expression and Immune Cell Infiltration in KIRC

Lastly, we analyzed the association between KRAB-ZNF expression and the immune landscape of KIRC tumors using previously estimated immune cells fractions. As presented in Figure 6A, both the leukocyte fraction and the LIS scores are negatively correlated with most KRAB-ZNFs (*n* = 22 and *n* = 16, respectively). Specifically, the abundances of regulatory T cells (Tregs), follicular helper T cells (Tfh), and gamma/delta T cells (γδ T cells) are negatively correlated with the levels of *n* = 20, *n* = 16, and *n* = 17 KRAB-ZNFs, respectively (Figure 6B). In contrast, the presence of resting NK cells and resting mast cells is significantly positively associated with the expression of most KRAB-NFs (*n* = 20 and *n* = 15, respectively). As for other immune cell subtypes, we observed both positive and negative associations, depending on the KRAB-ZNF. 

We also looked at the activation of immune-associated biological processes in KRAB-ZNF-related transcriptome profiles using the GSEA. As presented in Figure 6C, we observed significant depletion of inflammatory response (*n* = 14), IFN-α (*n* = 15) and IFN-γ response (*n* = 14), as well as IL2-STAT5 (*n* = 12), IL6-JAK-STAT3 (*n* = 13), and TNF-α signaling (*n* = 10), in a substantial number of KRAB-ZNF-associated transcriptome profiles followed by significant enrichment of the TGF-β-mediated signaling pathway (*n* = 13 KRAB-ZNF-associated transcriptome profiles). 

## 4. Discussion

To date, the role of KRAB-ZNFs in clear cell kidney carcinoma development and progression is largely unknown. Here, we demonstrate for the first time a significant association between the upregulation of specific KRAB-ZNFs with the better survival of KIRC patients, which strongly corresponds to tumor dedifferentiation status (mostly negatively) and the level of immune cell infiltration. We also revealed a potential prognostic role for the expression of several KRAB-ZNF and uncovered the association with cancer stemness-like phenotype. 

The role of most KRAB-ZNF transcription factors in solid tumors’ development and progression remains largely unknown, with only several (out of ~700 proteins) being well studied and documented as either tumor suppressors or oncogenes or harboring both tumor-promoting and -suppressing roles [2,5]. Moreover, the engagement of KRAB-ZNFs in the regulation of cancer stem cell acquisition and maintenance still remains unexplored. Similarly, the part for most KRAB-ZNFs in anticancer immune response persists undiscovered.

The canonical function of KRAB-ZNFs is to epigenetically repress specific regions within the chromatin, significantly impairing gene expression [3]. To exert their function, KRAB-ZNFs recruit KAP1 protein (also known as TRIM28 or TIF1β)—a scaffold protein facilitating the formation of a histone-modifying complex that consequently triggers heterochromatinization and target gene repression [4]. 

Recently, we have demonstrated that KAP1/TRIM28 is significantly associated with cancer stemness across distinct types of solid tumors [27,28,29]. Here, we focused on those KRAB-ZNF members with confirmed direct interaction with KAP1/TRIM28 protein as several of them might serve as the executors of KAP1/TRIM28-associated stem cell-like tumor phenotype. According to both ComPPI and Pathway Commons PPI [14,15], 28 KRAB-ZNFs interact with KAP1/TRIM28 directly. To date, there are no data regarding solid cancer development and progression for 12 of them, namely ZFP1 (ZNF475), ZNF2, ZNF74, ZNF140, ZNF197, ZNF256, ZNF274, ZNF620, ZNF585B, ZNF747, ZNF764, and ZNF829. This is the first report demonstrating a significant association of the abovementioned KRAB-ZNFs with KIRC patients’ survival (mainly with a better outcome), tumor dedifferentiation status (depletion of cancer stemness), and the landscape of immune cell infiltration. Moreover, using our approach, we were able to determine the prognostic value for ZNF2, ZNF256, ZNF585B, and ZNF829 (together with three other KRAB-ZNFs) in predicting KIRC patients’ outcomes. We suggest that, at least in KIRC tumors, these KRAB-ZNFs act as tumor-suppressive transcription factors.

As for other KRAB-ZNFs preselected in our study, several reports regarding tumor development and progression might be found, although with no relevance to clear cell kidney tumors. Briefly, ZFP57—the embryonic stem cell-specific transcription factor—is involved in colorectal cancer spreading [30], and its overexpression was observed in high-grade gliomas [31], suggesting the oncogenic role in tumorigenesis. On the other hand, ZFP57 was recently reported to suppress the proliferation of breast cancer cells through downregulation of the Wnt/β-catenin signaling pathway [32]. According to our data, ZFP57 is not associated with KIRC patients’ survival or with the dedifferentiation status of the tumor, suggesting that it might play a context-dependent role in tumorigenesis. 

ZNF10 promotes the carcinogenesis and progression of breast invasive ductal carcinoma via the Wnt/β-catenin signaling pathway [33] and recently was identified as a component of a radiotherapy-related four-gene signature that predicts the survival of head and neck squamous cell carcinoma patients [34]. Similar to ZFP57, our data demonstrate that the level of ZNF10 is not related to KIRC patients’ survival, although it is negatively associated with cancer stemness. 

ZNF195 is one of the most significant survival-associated genes in bladder cancer [35] and has been previously reported as a biomarker for gemcitabine sensitivity in head and neck squamous cell carcinoma [36]. Our results demonstrate that ZNF195 is significantly positively associated with higher KIRC grade and stage, and high ZNF195-expressing tumors possess cancer stemness characteristics, although the level of ZNF195 does not correlate with patient prognosis. 

Among the tested KRAB-ZNFs, numerous studies demonstrate significant engagement of ZNF224 in the regulation of cancer development and progression, especially in melanoma [37], breast cancer [38,39], and Wilms tumor [40]. However, the data for clear cell kidney carcinoma are still missing. In our results, ZNF224 is negatively associated with cancer dedifferentiation—KIRC tumors that exhibit cancer stem cell-like phenotype express significantly lower levels of ZNF224, suggesting its tumor-suppressive role in kidney cancer.

ZNF250 overexpression significantly predicted reduced survival of TCGA breast cancer patients [41]. The expression of ZNF324B, together with five other genes, could predict TCGA head and neck cancer patients’ prognosis [42]. The level of ZNF331 (with a panel of 14 other transcription factors) was recently identified with powerful predictive performance for the overall survival of hepatocellular carcinoma patients [43]. Moreover, the methylation status of ZNF331 is an independent prognostic marker of colorectal cancer [44]. ZNF331 exhibits both tumor-suppressive (in gastric cancer) and tumor-promoting activities (in colon cancer cells) [44,45]; however, its role in kidney cancer remains unknown. Here, we demonstrate that the overexpression of ZNF250, ZNF324B, and ZNF331 is significantly negatively associated with tumor dedifferentiation status and that higher-grade tumors (with cancer stemness characteristics) express substantially lower levels of those transcription factors. However, none of these markers possess prognostic value in KIRC patients. 

An increasing number of reports reveal the role of ZNF350 (ZBRK1) in suppressing the progression of distinct types of solid tumors, including cervical [46,47], breast [48], and kidney cancers [49]. In our data, ZNF350 expression is significantly negatively correlated with cancer stemness and is associated with the better overall survival of KIRC patients, suggesting its tumor-suppressive role in kidney tumors.

ZNF382 was firstly identified as a direct repressor for several oncogenes (i.e., MYC, MITF, HMGA2, and CDK6) across distinct types of cancers, including lung, esophageal, colon, stomach, breast, and cervical tumors [50]. Furthermore, ZNF382 serves as a tumor suppressor in gastric and liver cancers, exerting its function through the inhibition of EMT and Wnt/β-catenin pathways, respectively [51,52]. Recently, ZNF382 was reported as a component of an epigenetic signature that predicts the survival of laryngeal squamous cell carcinoma [53]. Moreover, with two other protein-coding genes, ZNF382 forms a prognostic signature that robustly predicts the outcome of bladder cancer patients [54]. Distinct methylation pattern of ZNF382 promoter might serve as a biomarker in pancreatic ductal carcinoma [55]. However, the role of ZNF382 in kidney cancer has not been determined yet. Our results clearly demonstrate the negative association between ZNF382 level and cancer stemness, which strongly corresponds to better overall survival of patients. Moreover, together with six other KRAB-ZNFs, ZNF382 possesses a significant prognostic value in KIRC tumors.

High expression of ZNF420, also known as APAK, is correlated with a poor prognosis of colorectal cancer patients [56]. Similarly, overexpression of ZNF460 predicts worse survival and promotes metastasis through JAK2/STAT3 signaling pathway in patients with colon cancer [57]. As for ZNF496, a recent report demonstrates that it acts as a target gene-specific ERα corepressor and inhibits the growth of breast cancer, suppressing the development of ERα-positive tumors [58]. Our data strongly suggest that ZNF420 and ZNF496 are both involved in KIRC progression as tumor suppressors, as their upregulation corresponds to better patient survival and lower tumor grade followed by a significant depletion of cancer stemness characteristics.

ZNF689, also known as TIPUH1, promotes the progression of hepatocellular carcinoma by suppressing the apoptotic signaling, and a high ZNF689 level indicates a poor prognosis of hepatocellular carcinoma [59,60]. Recently, ZNF689 was shown as a direct regulator of pancreatic cancer cell invasion and migration [61]. We suggest that the role of ZNF689 in tumorigenesis is highly context-dependent, as our results revealed a positive association with KIRC patients’ outcomes. High ZNF689 expression was observed in lower-grade tumors and was negatively correlated with cancer stemness traits. 

As cancer stemness is significantly negatively associated with anticancer immunity [62], we also looked at the level of immune cell infiltration in KIRC tumors. Surprisingly, the expression of most KRAB-ZNFs is also negatively associated with transcriptome-based immune scores, namely the leukocyte fraction estimate and lymphocyte infiltration signature score [25], suggesting that both infiltrating immune cells and cancer stem cell-like characteristics are depleted in KRAB-ZNF^HIGH^ tumors. 

Notably, we observed the negative correlation between the expression of most KRAB-ZNFs and the abundance of regulatory T cells (Tregs), follicular helper T cells (Tfh), and gamma/delta T (γδ T) cells. In cancer, the presence of Tregs prevents the development of effective antitumor immunity in tumor-bearing patients [63]. A recent finding suggests that Tregs might directly facilitate cancer stemness maintenance [64]. Therefore, it is not surprising that high KRAB-ZNF expression is associated with attenuated cancer stemness and better patient prognosis followed by depleted regulatory T cell infiltration level in kidney cancer patients. 

The γδ T cells might promote or suppress tumor progression (either directly or through indirect effects), and their actions are robustly influenced by the cytokines present in the tumor microenvironment. However, the role of γδ T cells in the regulation of cancer stemness is still unknown. Recent in vitro studies suggest that the cancer stem cell population exhibits strong resistance to γδ T cell-mediated killing. However, further experimental confirmation is needed to prove this phenomenon in vivo [65]. 

On the other hand, we observed a significant positive association between the expression of most KRAB-ZNFs and the abundance of resting NK cells or resting mast cells and observed a significant positive correlation with both naïve and memory CD4+ T cells for almost half of the tested KRAB-ZNFs. An increasing number of data demonstrate that NK cells can selectively identify and kill the population of dedifferentiated cancer cells that possess the characteristics of stem cells [66,67]. As for mast cells, they have both pro- and antitumorigenic roles depending on the cancer type, tumor progression level, and localization within the tumor [68]. Here, we demonstrate that high KRAB-ZNF-expressing KIRC tumors are significantly enriched with NK cells and mast cells while showing depletion of cancer stemness traits. However, further studies are needed to verify whether KRAB-ZNFs play a direct role in accumulating these immune cell populations within the tumor. 

Moreover, within the KRAB-ZNF-associated transcriptome profiles, we observed significant downregulation of genes expressed in response to interferons α and γ and a substantial depletion of signaling mediated by other proinflammatory cytokines: IL2, IL6, and TNF-α. 

IFNs participate in tumor immunology as a “double-edged sword”, exhibiting both pro- and antitumorigenic activities [69], and recent studies demonstrated their involvement in the regulation of cancer stem cell survival and metastatic progression in distinct types of solid tumors [70,71,72,73]. For both TNF-α and IL6 cytokines, their activity is highly complex and context-dependent. TNF-α was reported to directly increase the stem cell-like properties of cancer cells, specifically in the kidney [74], breast [75], and colon cancers [76] and melanoma [77]. IL6 also triggers enhanced stemness of cancer cells in solid tumors, promoting the self-renewal and preventing apoptosis of stem-like cells, especially in lung cancer [78], glioma [79], and colorectal cancer [80,81]. Recent data for clear cell renal cancer demonstrate that IL6 directly enhances tumor progression and stemness acquisition and allows cells to overcome natural tumor-suppressive mechanisms [82].

The abovementioned inflammatory pathways can enhance tumor growth and immune escape, especially by favoring cancer stemness. Therefore, a significant attenuation of this signaling in KRAB-ZNF^HIGH^ KIRC tumors might directly link to robust depletion of cancer stem cell-like traits and ultimately result in better patient outcomes. However, the exact role of KRAB-ZNFs in this scenario remains to be elucidated. As KRAB-ZNFs are potent transcriptional suppressors, it might be speculated that at least several of them directly target inflammatory-responsive genes and keep them repressed in kidney cancer in the absence of tumor-infiltrating lymphocytes. 

Lastly, we observed significant enhancement of the TGF-β mediated signaling pathway in high KRAB-ZNF-expressing tumors that do not exhibit cancer stemness phenotype. The TGF-β pathway is an important way to induce cancer stem cell formation in the epithelial cells of the lung [83], breast [84], colorectal [85], gastric [86], and kidney cancers [87], among others. Therefore, the upregulation of TGF-β signaling observed in KRAB-ZNF^HIGH^ KIRC tumors is surprising and requires further confirmation to delineate the impact on tumor dedifferentiation status. 

Using our KRAB-ZNF-based gene signature, we were able to filter out high-risk KIRC patients that exhibit stemness-high phenotype accompanied with an enriched immune-suppressive microenvironment that collectively leads to worse patient outcomes. Our results provide new insight into the role of TRIM28-interacting KRAB-ZNF proteins in kidney cancer development.

## 5. Conclusions

To date, our report is the first that investigates the role of TRIM28-interacting KRAB-ZNFs in kidney cancer progression. Our results demonstrate a robust negative association between KRAB-ZNFs and kidney cancer dedifferentiation status followed by attenuated proinflammatory signaling within the tumor. However, the exact mechanism mediating this stemness-low phenotype of KRAB-ZNF-overexpressing KIRC tumors should be confirmed experimentally. 

Moreover, we demonstrate for the first time the prognostic role for several KRAB-ZNFs in kidney cancer and believe that our findings may help better understand the molecular basis of KIRC. Ultimately, this approach may pave the way to more suitable prognostic tools for KIRC and promote the development of novel therapeutic strategies directly eradicating the dedifferentiated compartment of the tumor.

## Figures and Tables

**Figure 1 cancers-13-04835-f001:**
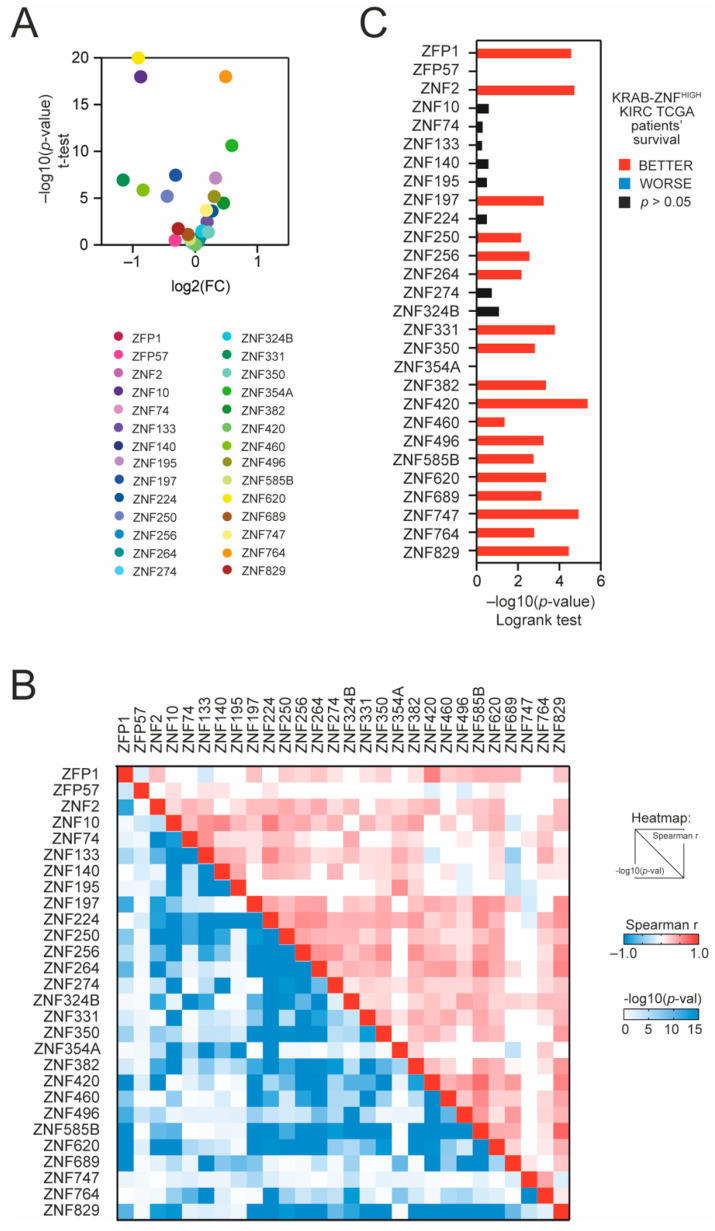
The expression of TRIM28-interacting KRAB-ZNFs in tumor and normal adjacent tissues and the association with kidney renal clear cell carcinoma (KIRC) patients’ survival. (**A**) The expression of TRIM28-interacting KRAB-ZNFs in tumor tissues and normal adjacent tissues based on TCGA data. (**B**) Correlation of KRAB-ZNF expression in KIRC tumor samples. The upper part of the heatmap presents the Spearman correlation coefficient, while the lower part of the heatmap shows the statistical significance of the correlation (−log10-transformed). (**C**) The hazard ratio (log10(HR)) of death for patients with high expression of specific KRAB-ZNFs (with the mean as a cut-off). Red and blue denote lower and higher hazard ratios, respectively.

**Figure 2 cancers-13-04835-f002:**
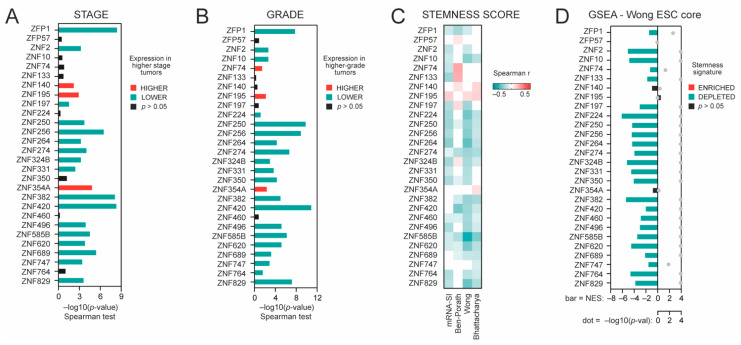
The association between KRAB-ZNF expression and tumor dedifferentiation status. (**A**) The association between KRAB-ZNF expression and tumor stage—either lower (green) or higher stage (red), as determined with Spearman correlation test (−log10(*p*-value)). (**B**) The association between KRAB-ZNF expression and tumor grade—either lower (green) or higher grade (red), as determined with Spearman correlation test (−log10(*p*-value)). (**C**) The heatmap of Spearman’s correlation between KRAB-ZNFs’ expression and four distinct stemness indices (mRNA-SI, Ben-Porath signature, Wong signature, Bhattacharya signature). Red and green denote positive and negative correlation, respectively. Only statistically significant associations are shown (*p* < 0.05). (**D**) The Gene Set Enrichment Analysis (GSEA) using genes significantly correlated (FDR < 0.01) with KRAB-ZNF level in KIRC patients was performed with the stemness signature (Wong_ESC_Core) as a reference. Bar—the normalized enrichment score (NES). Grey dot—statistical significance (−log10FDR).

**Figure 3 cancers-13-04835-f003:**
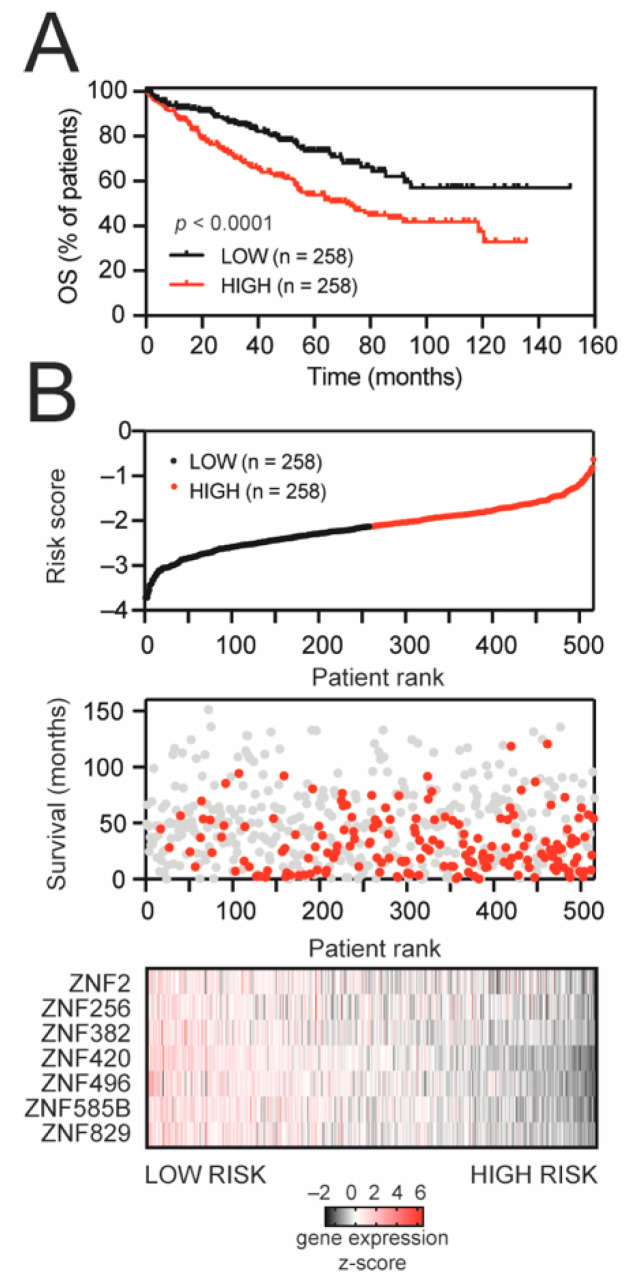
KRAB-ZNF-based gene signature for survival prediction in KIRC patients. (**A**) Kaplan–Meier survival curves for the entire TCGA KIRC set, stratified into high-risk and low-risk groups based on median value of developed risk score. Red and black denote high- and low-risk patients, respectively. (**B**) The signature-based risk score distribution (upper panel: red—high risk, black—low risk), patients’ survival status (middle panel: red—dead, grey—alive), and heatmap of seven KRAB-ZNF expression profiles (lower panel: black—downregulated, red—upregulated expression) in the entire KIRC set.

**Figure 4 cancers-13-04835-f004:**
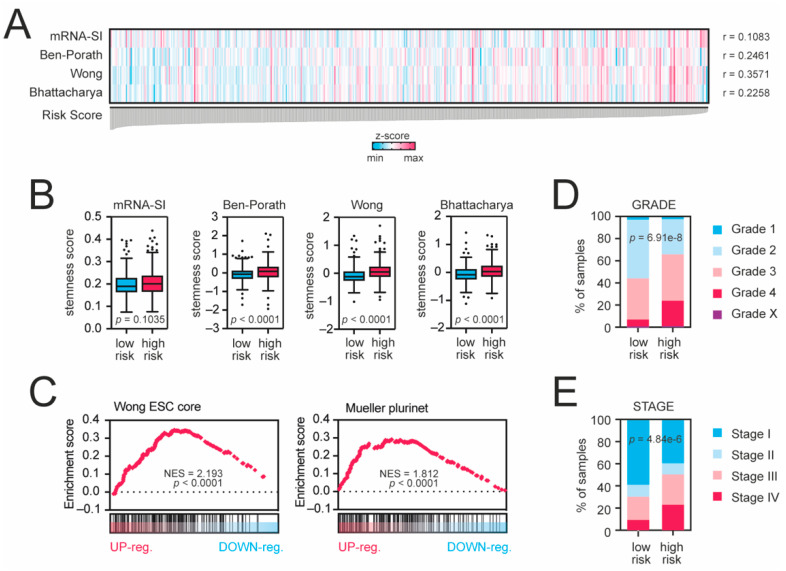
Tumors from high-risk patients exhibit cancer stem cell-like traits. (**A**) KRAB-ZNF-based risk score is significantly correlated with cancer stemness scores. All samples were ranked according to the risk score value. Blue and magenta present lower and higher stemness scores, respectively (z-score-transformed). (**B**) Tukey box plots presenting the levels of stemness scores in low-risk (blue) and high-risk (magenta) KIRC patients. (**C**) The transcriptome profiles of high-risk KIRC patients are significantly enriched with stemness markers. The GSEA using all significantly differentially expressed genes (DEGs with *p* < 0.05 and FDR < 0.05) in high-risk KIRC patients was performed with the stemness signatures “Wong_ESC_core” or “Mueller_Plurinet” as references. (**D**) Tumors from high-risk patients are higher-grade tumors. Tumor grades are color-coded as presented in the legend. (**E**) Among the high-risk patients, the frequency of higher stage is more significantly elevated. Tumor stages are color-coded as presented in the legend.

**Figure 5 cancers-13-04835-f005:**
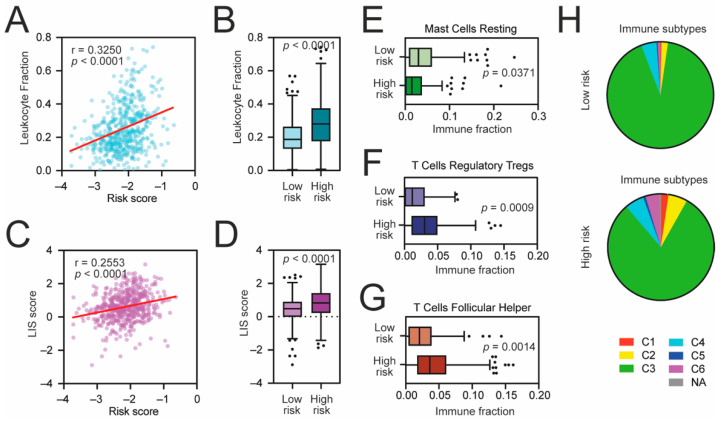
The immune landscape of high-risk KIRC tumors. (**A**) The Spearman correlation of the leukocyte fraction and KRAB-ZNF-based risk score in TCGA KIRC patients. (**B**) Tukey box plots of the leukocyte fraction in low-risk and high-risk KIRC patients. (**C**) The Spearman correlation of the lymphocyte infiltration signature (LIS) score and KRAB-ZNF-based risk score in TCGA KIRC patients. (**D**) Tukey box plots of the LIS score in low-risk and high-risk KIRC patients. (**E**–**G**) Tukey box plots of CIBERSORT-estimated fractions of (**E**) resting mast cells, (**F**) regulatory T cells, and (**G**) follicular helper T cells in low-risk and high-risk KIRC patients. (**H**) Classification of low-risk and high-risk KIRC tumors into C1–C6 immune subtypes [25]. Red—C1, wound healing; yellow—C2, IFN-γ dominant; green—C3, inflammatory; light Blue—C4, lymphocyte depleted; dark blue—C5, immunologically quiet; purple—C6, TGF-β dominant; Grey—not applicable.

**Figure 6 cancers-13-04835-f006:**
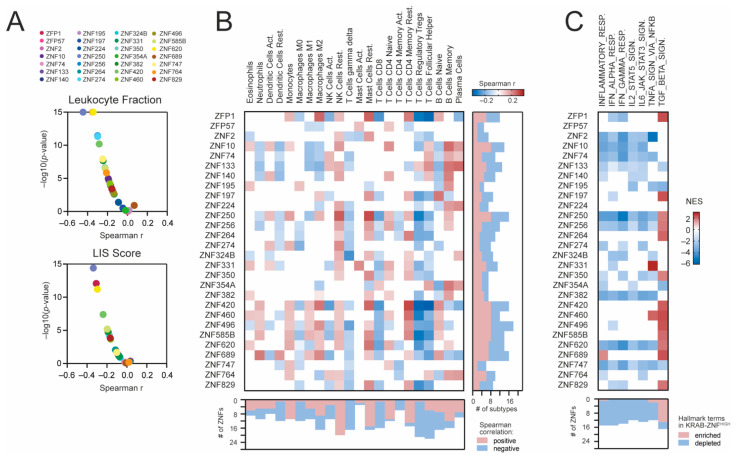
The association between KRAB-ZNF expression and the immune cell infiltration in KIRC. (**A**) The Spearman correlation of the leukocyte fraction (upper panel) or the lymphocyte infiltration signature (LIS) score (bottom panel) and KRAB-ZNF expression in TCGA KIRC patients. (**B**) The Spearman correlation of immune cell infiltration level with the expression of KRAB-ZNFs in KIRC. Blue and brown present negative and positive correlations, respectively. The right panel denotes the number of immune cell subtypes positively (brown) or negatively (blue) associated with the expression of distinct KRAB-ZNFs. The bottom panel denotes the number of KRAB-ZNFs positively (brown) or negatively (blue) associated with the abundance of distinct immune cell subtypes. (**C**) The heatmap presents the normalized enrichment scores (NES) for 7 selected immune-associated Hallmark terms (v7.4) from the GSEA analysis of KRAB-ZNF-associated transcriptome profiles. White—no statistical significance (*p* > 0.05) or no DEGs detected. The bottom panel denotes the number of KRAB-ZNF-associated transcriptome profiles with either enriched (brown) or depleted (blue) immune-associated terms.

## Data Availability

The data that support the findings of this study are openly available in TCGA at https://portal.gdc.cancer.gov/ (accessed on 15 July 2021) and in GEO at https://www.ncbi.nlm.nih.gov/geo/ (accessed on 15 July 2021). Access to both databases is unrestricted and does not require patients’ consent or other permissions. The use of the data does not violate the rights of any person or any institution.

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
