# Peer review of "Low Levels of TRIM28-Interacting KRAB-ZNF Genes Associate with Cancer Stemness and Predict Poor Prognosis of Kidney Renal Clear Cell Carcinoma Patients"

_cancers, 2021, doi:10.3390/cancers13194835_

Round 1

Reviewer 1 Report

Good manuscript on “Low levels of TRIM28-interacting KRAB-ZNF genes associate with cancer stemness and predict poor prognosis of Kidney Renal Clear Cell Carcinoma patients” where authors showed the involvement of TRIM28-interacting KRAB-ZNFs in kidney cancer progression. Authors further showed negative association between KRAB-ZNFs and cancer stemness and an attenuated immune-suppressive response which reveal the prognostic role for several KRAB-ZNFs.

The authors should check for the spelling in the text, and grammatical error, there are many mistakes.

Reviewer 2 Report

In this study, the authors attempt to establish that low levels of TRIM28-interacting KRAB-ZNF genes associate with cancer stemness and serves as an indicator of poor prognosis of Kidney Renal Clear Cell Carcinoma (KIRC) patients.

The overall paper nicely describes the sufficient background TRIM28-interacting KRAB-ZNF with their molecular function. The methods are sufficiently described and results are presented with the necessary statistical supports for significant findings.

The study for the first time reports the negative association of TRIM28-interacting KRAB-ZNF genes with cancer stemness, and kidney cancer de-differentiation status followed by attenuated proinflammatory signaling within the tumor.

The results from survival analyses highly support the prognostic value of several KRAB-ZNFs in KIRC. 

The findings of the paper also motivate future experimental validations to elucidate the underlying KRAB-ZNFs related molecular mechanisms driving effect on KIRC.

Comments

Figure 1A. The volcano plot with log-fold changes in expression (tumor vs normal) on X-axis and P-values on Y-axis would give a better representation of the data.

Figure 1B, 2C, 3B, 6B, 6C. The color gradients on scales for p and r values are missing.

The figures in supplementary data should be immediately followed by appropriate captions/legends instead of a separate section of legends.
